# Declining harbour seal abundance in a previously recovering meta-population

**Daire Carroll** [1,2]*, **Markus P. Ahola** [3,4], **Anja M. Carlsson** [3], **Anders Galatius** [5],
**Kjell T. Nilssen** [6], **Tero Härkönen** [7], **Karin C. Harding** [1,2]

**1** Department of Biological and Environmental Sciences, University of Gothenburg, Gothenburg, Sweden,
**2** Gothenburg Global Biodiversity Centre, Gothenburg, Sweden, **3** Department of Population Analysis and
Monitoring, Swedish Museum of Natural History, Stockholm, Sweden, **4** Marine Environment Research
Group, Turku University of Applied Sciences, Turku, Finland, **5** Section for Marine Mammal Research,
Department of Ecoscience, Aarhus University, Roskilde, Denmark, **6** Institute of Marine Research,
Tromsø, Norway, **7** Maritimas AB, Kärna, Sweden

* daire.carroll@bioenv.gu.se

journal.pone.0326933

Bari Aldo Moro, ITALY

**Peer Review History:** PLOS recognizes the
benefits of transparency in the peer review
process; therefore, we enable the publication
of all of the content of peer review and
author responses alongside final, published
articles. The editorial history of this article is
available here: https://doi.org/10.1371/journal.
pone.0326933

## Abstract

We present evidence that the abundance of harbour seals in the Kattegat-Skagerrak
is in decline. Until recently, the Kattegat-Skagerrak harbour seal population has
grown exponentially as it has recovered from historic over-hunting and two mass
mortality events. This has provided an important case study for the influence of
environmental factors on population growth. Over recent years, deteriorating aver-
age body condition and reduced pup counts in certain colonies have indicated that
the population is under stress. At the same time, there has been an increase in
hunting in the region which may compound existing environmental stressors. To
determine trends in harbour seal abundance, we compile aerial survey data for the
Kattegat-Skagerrak and South-Western Baltic Sea (S.W. Baltic) populations between
the years 2003 and 2023. With parametric modelling, we find that a logistic growth
model is the best fit for aerial survey data from the Kattegat-Skagerrak population,
while an exponential growth model is the best fit for data from the smaller S.W.
Baltic population. We determine trends by fitting non-parametric Generalised Addi-
tive Models (GAMs) to aerial survey data and calculating their first derivative. In the
Kattegat-Skagerrak, we estimate an annual decline of $-408$ individuals (SE = 242,
$CI_{95\%}$ = [$-882, 67$]) at the end of the survey period. This decline represents approxi-
mately 3.3% of the estimated mean count in 2023. Rate of change remained positive
in the separate S.W. Baltic population throughout the survey period at an estimated
annual increase of 54 individuals (SE = 23, $CI_{95\%}$ = [9,99]), representing approxi-
mately 3.9% of mean counts in 2023. We find that the final rate of change was zero
or below zero in eleven out of twelve subregions (representing individual colonies) in
the Kattegat-Skagerrak. Declining counts in the Kattegat-Skagerrak since the mid-
2010s may be influenced by changes in haul-out behaviour as a result of increased
population density or vessel traffic. Despite this, declines likely reflect real changes

**Data availability statement:** All data and code are available on GitHub: https://github.com/DaireCarroll2023/PV_Trend_Analysis and Zenodo DOI: https://doi.org/10.5281/zenodo.14945453.

**Funding:** Swedish surveys were funded by the Swedish Environmental Protection Agency, and the Swedish Agency for Marine and Water Management. Danish surveys were funded by the Danish Environmental Protection Agency. DC thanks the Wild Animal Initiative Fellowship, grant number F-2023-00005, and the Swedish Environmental Protection Agency through the Environmental Research Fund and the Swedish Research Council Formas, grant number 2024-00147. KCH was supported by the European Union MARHAB, grant no. 101135307, and the Swedish Research Council Formas, grant no. 2023-00502. There was no additional external funding received for this study.

**Competing interests:** The authors have declared that no competing interests exist.

in seal abundance. Changes in abundance may be the result of well documented environmental degradation affecting prey availability and causing regionally lowered birth rates.

## Introduction

Bioindicator species are monitored in order to track changes in environmental state [1]. European seal populations have a long history of use as bioindicators due to their responsiveness to environmental factors such as prey abundance and pollution [2,3]. Harbour seals (*Phoca vitulina*) in the Kattegat-Skagerrak have been the focus of extensive ecological monitoring over the past four decades [4–6]. Continued monitoring of the population's response to ecological change provides an opportunity to investigate the processes affecting trends in marine mammal abundance.

The Kattegat-Skagerrak harbour seal population is largely isolated from the closest populations in the South-Western Baltic Sea (S.W. Baltic), Limfjord, and Wadden Sea [7]. Paralleling many other pinniped populations, the Kattegat-Skagerrak population declined rapidly in the early half of the twentieth century as a result of culling [4,8,9]. Sustained hunting through the middle of the century led to further decline and prevented recovery [5,10]. Harbour seals were protected from hunting in 1967 in Sweden and 1976 in Denmark [4,5]. Following this, the population grew exponentially through the 1980s and 1990s [5]. Mass mortality events, caused by outbreaks of Phocine Distemper Virus (PDV) in 1988 and 2002, killed up to 60% of the population on each occasion [11,12].

The highest possible intrinsic annual growth rate for a harbour seal population with a stable age structure is slightly less than 13% [13]. Following the 1988 PDV outbreak, the Kattegat-Skagerrak population displayed exponential growth of between 10.2% and 13.6% annually [5]. Temporary rates of increase above 13% were likely the result of the population structure being skewed towards young females following the mass mortality event [5,13,14]. This period of exponential growth set a baseline for the conditions of fecundity and survival which can be expected in an optimal environment without mass mortality events or density dependence. Prior to the 2002 PDV outbreak, growth rates in the Kattegat-Skagerrak began to slow with mean counts of approximately 10,000 individuals [5]. Although the population continued to grow following the 2002 PDV outbreak, growth rates were lower than in the previous decades, at 5% and 6.5% for the Kattegat and Skagerrak respectively [14]. These rates were influenced by increased mortality during 2007, which was potentially caused by exposure to algal toxins and bacterial infections, and in 2014, caused by an outbreak of influenza A(H10N7) [15–17]. Mortality during these events, however, was considerably lower than during either of the two PDV outbreaks and they do not explain continued low growth rates on their own [5,14]. This has led to the hypothesis that the population may be experiencing resource limitation, which is further supported by observed reductions in somatic growth rates and reduced pup counts in the Skagerrak, although trends may differ elsewhere [14,18,19].

Changes in harbour seal abundance and growth rates can be driven by a variety of natural (e.g., predation and intra- or inter-specific competition) and anthropogenic (e.g., hunting, pollution, and bycatch) factors [3,4,20–25]. Recent decades have seen wide-scale changes in the Kattegat-Skagerrak ecosystem. The area has been the subject of intensive industrial fishing since the 1970s, leading to severe declines in many fish populations [26–28]. Species which once formed a large component of seal diets, such as herring (*Clupea harengus*) and Atlantic cod (*Gadus morhua*), are now less common [26,29–31]. Declines in the abundance of large predatory fish, such as Atlantic cod, have led to a large-scale ecological shift with smaller fish now dominating the ecosystem alongside abundant filamentous algae [32,33]. Harbour seals are opportunistic feeders [29,30,34] and have likely shifted their diet in response to these changes, potentially limiting population growth. In combination with these ecological shifts, runoff from agriculture and forestry has led to wide-scale eutrophication and brownification [35–37]. There has also been an increase in the hunting of harbour seals, particularly in Sweden, over the past decade [14]. As harbour seals exhibit high levels of site fidelity, subpopulations (often referred to as colonies) are impacted by local environmental conditions and anthropogenic stressors as well as local management strategies which may result in variable growth rates [7,21,38,39].

Given wide-scale changes in the Kattegat-Skagerrak ecosystem and increased hunting, it is imperative for the continued use of the species as a bioindicator that changes in abundance are documented. To this end, we have compiled data from harbour seal moult surveys carried out throughout the Kattegat-Skagerrak as well as the nearby S.W. Baltic population since 2003. Through a combination of parametric and non-parametric modelling, we estimate trends in harbour seal counts for the combined Kattegat-Skagerrak and for subregions representing seal colonies. This work presents a valuable case study of potential growth limitation for a marine mammal population and an opportunity to begin to disentangle the influence of natural processes from human driven environmental change. The results are highly relevant for the management of marine wildlife.

## Methods

### Data collection

The time period between 2003 and 2023 was considered, representing the years since the most recent outbreak of PDV in 2002 [12]. Data were compiled for the Kattegat-Skagerrak and S.W Baltic (Fig 1). The Kattegat-Skagerrak can be considered a metapopulation with limited exchange of individuals between subpopulations (colonies) [4,7,13]. However, for management purposes, the Kattegat and Skagerrak are often considered separately [6]. The S.W. Baltic was considered as a separate population based on low levels of exchange evident from genetic analysis, telemetry studies, and patterns of spread during disease outbreaks [7,16,38].

We divided the study area into three sea regions (the Kattegat, the Skagerrak, and the S.W. Baltic) and 18 subregions, representing colonies (Fig 1) [4,7,14]. The structuring of harbour seal populations into colonies is well documented [7,39]. The exact definition of colony boundaries, however, is not fully resolved for the Kattegat-Skagerrak. Although harbour seals exhibit a high degree of site fidelity during the breeding season, they may display more migratory behaviour during the moulting season [21,40]. In our analysis, subregion boundaries were based on previous work in order to facilitate comparison and explore local trends in seal abundance [4,14].

Aerial surveys were conducted during the peak in haul-out numbers coinciding with the annual moult in late August. In all cases, surveys were carried out on days with no precipitation and windspeeds of less than $10\,\mathrm{ms^{-1}}$ [41].

One Norwegian subregion (subregion 5) was included in Swedish surveys. In Sweden (subregions 5–12 and parts of 17) and Denmark (subregions 13–16 and parts of 17), haul-outs were photographed from light aircrafts [4,41]. Individual surveys of each region took place on a single day between the hours of 08:00 and 15:00. Each haul-out was surveyed between two and three times each year, except for 2004, when Danish haul-outs were only surveyed once.

In Norway (subregions 1–4), haul-outs were photographed using drones flown from a boat at a distance of between 800 and 1000 m. Surveys were carried out during daylight hours within two hours of the first low tide of the day. Individual

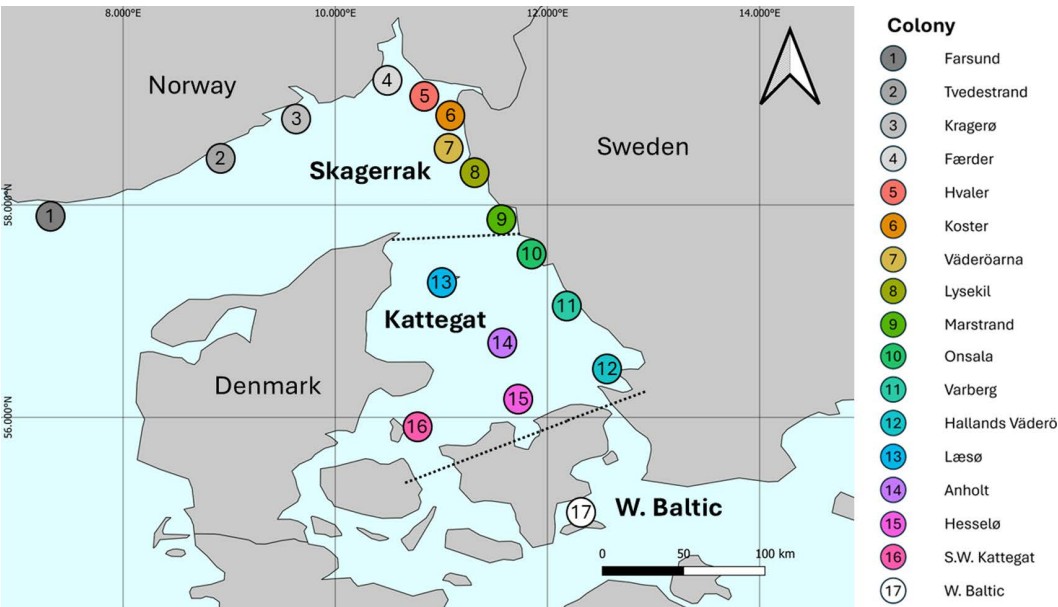

**Fig 1. Map of the survey area.** Subregions 1 to 8 belong to the Skagerrak region. As data for subregions 1 to 4 were only available for 2016 and 2022, they were excluded from statistical analysis of trends. Subregions 9 to 16 belong to the Kattegat region. The S.W. Baltic (17) was treated as a separate population with just one subregion. Administrative boundary polygons sourced from Natural Earth.

subregions were surveyed on a single day. Each subregion was surveyed between one and three times on different days in 2016 and again in 2022.

Seals were then counted from images. Two independent observers counted each image. If the count for a location differed by more than 5% between observers, another count was carried out for that location until there was agreement between counts [42]. For each region and each subregion, the sum of counts for each survey day was calculated. As data for only two years were available for subregions 1–4, they were excluded from counts for the Skagerrak region and all subsequent statistical analysis of trends. Previous analysis has indicated that mean daily count (mean count) is more effective for assessment of trends in harbour seal abundance than other metrics, such as maximum daily count [41]. For each year for which more than one count was available, mean counts were determined for all regions and subregions.

### Analysis of trends

**Parametric modelling.** Exponential growth of the Kattegat-Skagerrak harbour seal population was observed following their protection from hunting in the 1970s and following the outbreak of PDV in 1988 [4,5,14]. Given indications of declining growth rate over recent years, two classic parametric models for population growth were fit to mean count data ($N$) using a Non-linear Least Squares (NLS) regression. The first model represented exponential growth:

$$N_t = N_0 e^{rt}, \tag{1}$$

where $N_t$ is mean count at time $t$, $N_0$ is initial mean count (in 2003), and $r$ is the annual growth rate. The second model represented logistic (density dependent) growth:

$$N_t = \frac{K}{1 + \left(\frac{K - N_0}{N_0}\right) e^{-\mu t}}, \tag{2}$$

where $K$ is carrying capacity and $\mu$ is the intrinsic growth rate. Separate models were fit to data summarised at different spatial scales: **i.** for the combined Kattegat-Skagerrak and **ii.** for each region (the Kattegat, the Skagerrak, and the S.W. Baltic), taking "Year" as the explanatory variable and mean count ("Count") as response variable.

**Non-parametric modelling.** Parametric models (e.g., Equations 1 and 2) assume that population growth can be described by a fixed number of parameters (e.g., an intrinsic growth rate, carrying capacity, and initial population size). While they are informative when considering the underlying mechanisms controlling population growth, they also risk oversimplifying trends by, for example, assuming exponential, or density dependent growth with a fixed carrying capacity [43]. Under such assumptions, population decline following a period of population growth is unlikely to be detected. To explore trends without assuming exponential or logistic growth and to compare trends between regions and subregions, non-parametric Generalised Additive Models (GAMs) were fit to mean count data [44,45]. Separate models were fit to population counts, summarised at three different spatial scales, taking mean count ("Count") as the response variable and "Year" as a smooth term for: **i.** the combined Kattegat-Skagerrak, **ii.** the three regions (the Kattegat, the Skagerrak, and the S.W. Baltic, including "Region" as a smooth term and an interaction between "Year" and "Region"), and **iii.** for subregions in the Kattegat-Skagerrak (including "Subregion" as a smooth term and an interaction between "Year" and "Subregion"). Rates of change were determined by calculating the first derivative of fitted GAMs.

### Software and analysis

All analysis was carried out using the software R (version 4.3.2) [46] with organisation of data using the *tidyverse* package [47]. Analysis of spatial data and sorting of georeferenced observations into subregions was carried out using the *sf* package [48]. NLS regression was carried out using the *nls()* command with comparison of parametric model fit based on Akaike Information Criterion (AIC). GAM fitting was carried out using the *mgcv* package using method '*REML*', four initial basis functions (to prevent overfitting), and the *tp* (thin plate regression splines) smoother within smooth terms [49] with model predictions and determination of 95% Confidence Intervals ($CI_{95\%}$) made using the *investr* package [50] and estimation of first derivatives, Standard Errors (SEs), and $CI_{95\%}$ of derivatives using the *gratia* package [51].

Code is available on GitHub: https://github.com/DaireCarroll2023/PV_Trend_Analysis and Zenodo https://doi.org/10.5281/zenodo.14945453.

## Results

The results of aerial surveys were summarised to determine mean counts. Excluding the four Norwegian subregions surveyed in 2016 and 2022 only (subregions 1–4, Fig 1, S1 Table) and Danish subregions in the Kattegat in the year 2004, all subregions were surveyed each year.

### Analysis of trends

For count data pooled for the combined Kattegat-Skagerrak, as well as when treating data separately for the Kattegat and Skagerrak regions, the logistic model was found to be the best fit for mean count data based on lowest AIC. For data from the S.W. Baltic, the logistic model could not be fit, leading to the conclusion that the exponential model was the best fit (S2 Table, S1 Fig).

Non-parametric GAMs were fit to mean count data for the combined Kattegat-Skagerrak (adjusted $R^2 = 0.7$) as well as separately for the three regions (the Kattegat, the Skagerrak, and the S.W. Baltic, adjusted $R^2 = 0.96$) including "Region" as a smooth term and an interaction between "Year" and "Region" (Table 1, Fig 2A).

Estimated counts for the Kattegat and combined Kattegat-Skagerrak based on GAMs reached a maximum in 2017. Estimated counts for the Skagerrak reached a maximum in 2016. Estimates for the S.W. Baltic continued to increase throughout the survey period (S3 Table, Fig 2A).

**Table 1. Outcomes of GAM fitting for the combined Kattegat-Skagerrak population and including a main effect and interaction of region.**

| Main effects | | | | | | | |
|---|---|---|---|---|---|---|---|
| Model | Adj. R² | | | Intercept | SE | t-value | p-value |
| Kattegat-Skagerrak | 0.7 | | | 12,000 | 340 | 35.25 | < 0.001 |
| Regions | 0.96 | Kattegat | | 7,979 | 144 | 55.56 | < 0.001 |
| | | Skagerrak | | 3,827 | 200.4 | − 20.49 | < 0.001 |
| | | S.W. Baltic | | 855 | 200.4 | − 35.55 | < 0.001 |
| **Approximate significance of smooth terms** | | | | | | | |
| Model | Adj. R² | | | EDF | Ref. DF | *f*-value | *p*-value |
| Kattegat-Skagerrak | 0.7 | Year | | 2.43 | 2.78 | 16.63 | < 0.001 |
| Regions | 0.96 | Year | | 0.75 | 0.75 | 2.049 | 0.216 |
| | | Year*Kattegat | | 2.42 | 2.67 | 15.63 | < 0.001 |
| | | Year*Skagerrak | | 2.23 | 2.56 | 7.81 | < 0.001 |
| | | Year*S.W. Baltic | | 0.75 | 0.75 | 0.18 | 0.713 |

An asterisk (*) indicates an interaction. Adj. R² = adjusted R². SE = standard error. EDF = effective degrees of freedom. Ref. DF = reference degrees of freedom.

Rates of change were determined based on the first derivative of fitted GAMs (Fig 2B). These rates give a predicted change in count at any given time point. The final rate of change for the Kattegat-Skagerrak population at the end of the survey period was estimated to be – 408 (SE = 242, $CI_{95\%}$ = [- 882, 67]) individuals per year. A final rate of change of – 236 (SE = 111, $CI_{95\%}$ = [- 17, – 446]) was estimated in the Kattegat while – 228 (SE = 102, $CI_{95\%}$ = [- 430, – 27]) was estimated in the Skagerrak. In the S.W. Baltic population, a consistent annual increase of 54 individuals (SE = 23, $CI_{95\%}$ = [9,99]) was estimated across the entire survey period.

GAMs were also fit to count data for individual colonies including "Subregion" as a smooth term and an interaction between "Year" and "Subregion" (adjusted $R^2$ = 0.79, S4 Table, Fig 3). Rates of change of individual subregions were also determined (S2 Fig). Only one subregion (subregion 5) displayed a positive final rate of change while the remaining eleven displayed either a negative trend or little to no increase in mean count.

## Discussion

We have summarised the two most recent decades of harbour seal aerial survey data for the Kattegat-Skagerrak and S.W. Baltic. Counts in both the Kattegat and the Skagerrak reached a maximum around 2017 and have since declined. This contrasts with historical exponential growth observed between the 1980s and early 2000s [5]. The estimated count in 2023 for the combined Kattegat-Skagerrak (12,507, $CI_{95\%}$ = [11,558, 13,456]) was higher than the mean counts of approximately 10,000 individuals prior to the 2002 PDV outbreak [5]. Heide-Jorgensen and Härkönen (1988) estimated the size of the Kattegat-Skagerrak harbour seal population in 1890 to number 16,500 based on bounty statistics [4,5]. At the time, the population was already under considerable hunting pressure [4]. Although they should be considered as coarse estimates only, previous work has assumed the hauled out fraction of the Kattegat-Skagerrak harbour seal population observed during moult surveys to be between 57% and 65% [5,14]. Assuming either value would indicate that the Kattegat-Skagerrak harbour seal population in 2023 was close to 20,000, considerably larger than the 1890 estimate.

Contrastingly, counts continue to increase for the isolated S.W. Baltic population. Relative to the Kattegat-Skagerrak, counts in the S.W. Baltic in the 1970s were low (around 100). Hunting pressure was also high relative to the population size in the years leading up to the hunting ban [5,52]. As a result of their relatively slow intrinsic population growth rate, harbour seal populations today are still recovering from over-hunting in the previous century [42,53]. A probable explanation for the continued growth of the S.W. Baltic population is that it was close to extinction when protected from hunting

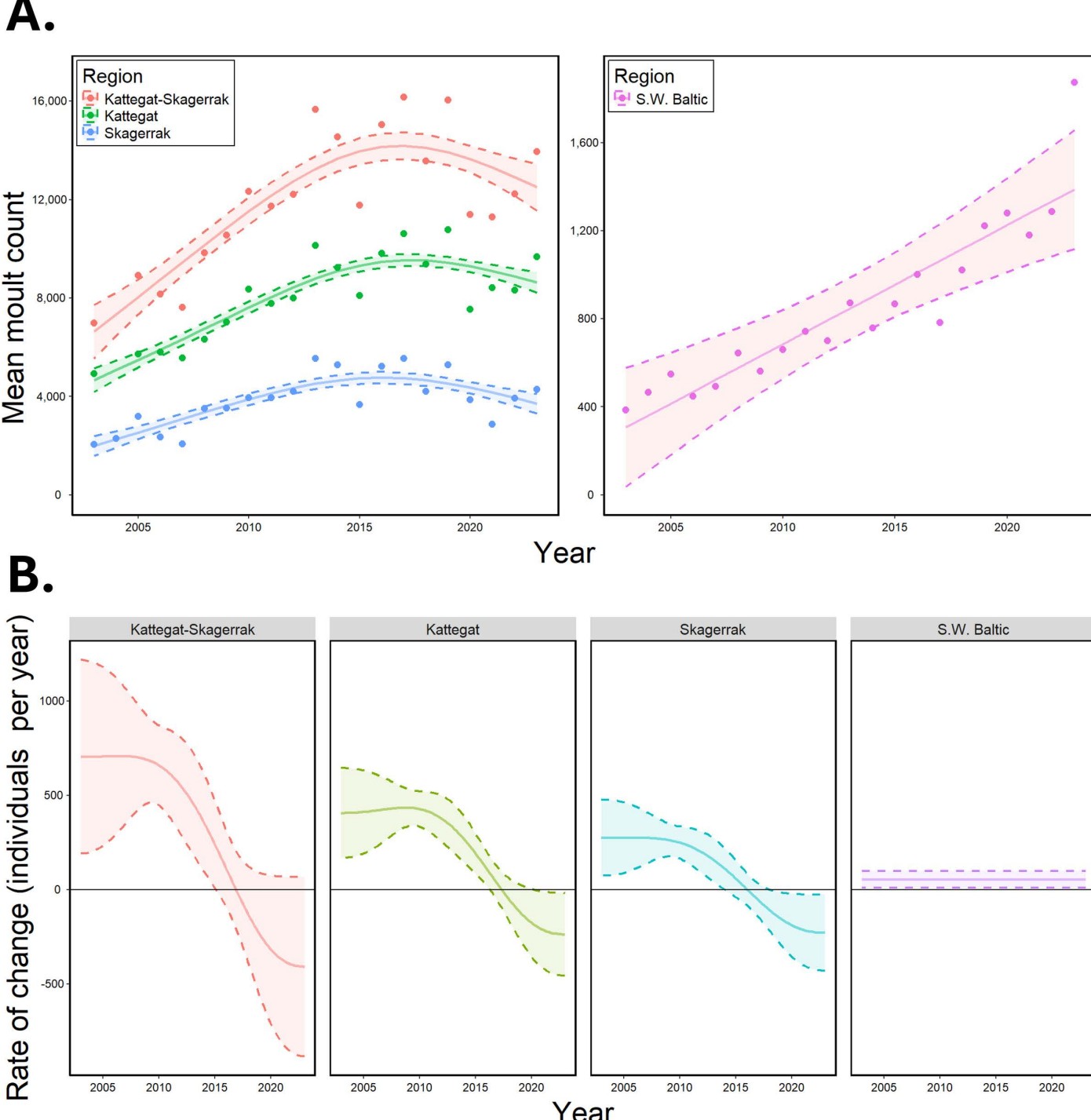

**Fig 2. Non-parametric modelling. A.** Mean number of harbour seals counted during moult surveys for different regions. Separate GAMs were fit to mean counts (points) for the combined Kattegat-Skagerrak (pink), and the three regions (the Kattegat: green, the Skagerrak: blue, and the S.W. Baltic: purple). Note the difference in y-axis scale between panels. **B.** Rates of change in population size were estimated based on the first derivative of fitted GAMs. Dashed lines represent $CI_{95\%}$ of estimates. Solid black lines are placed at zero, representing no growth.

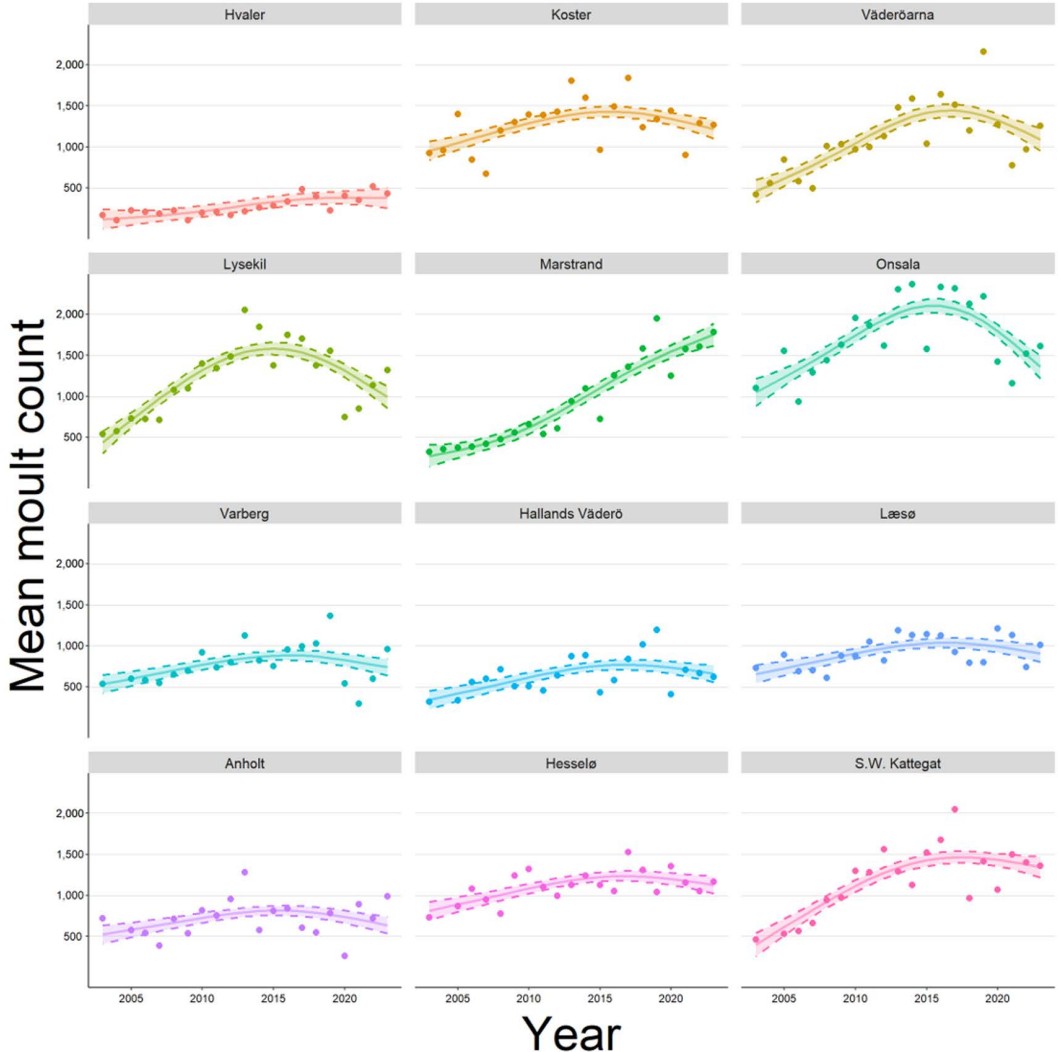

**Fig 3. Mean number of harbour seals counted during moult surveys for subregions in the Kattegat-Skagerrak metapopulation over the last 20 years.** Note an apparent change from exponential growth to linear growth and decline has occurred. A GAM was fit to mean counts (points) including subregion as a smooth term. Dashed lines represent CI$_{95\%}$ of estimates. See S4 Table for details on model fitting.

and that population density hence remains low relative to the Kattegat-Skagerrak. Environmental conditions may also differ between the two regions.

Harbour seals show high levels of breeding site fidelity and it is unlikely that significant levels of emigration out of the Kattegat-Skagerrak metapopulation have occurred over the past twenty years [7,38]. This is especially true considering the lack of growth in the nearby Wadden Sea population and the relatively small sizes of the closest populations in the Baltic Sea and Limfjord [5,53]. Limited migration does likely occur between individual colonies in the Kattegat-Skagerrak [39].

Over recent years, harbour seals have recolonised the South Funen Archipelago and Little Belt area of the Danish Straits, with average moult counts of 155 seals between 2021 and 2023 [42]. Four Norwegian subregions (Farsund, Tvedestrand, Kragerø, and Farsund) showed an increase in total mean count from 365 to approximately 658 between 2016 and 2022. The long interval between surveys in Norway prevented subsequent analysis of trends. It is possible

that increases along the Norwegian East Coast are driven by movement from the more densely populated Swedish West Coast. Although counts in both the Norwegian colonies and the South Funen Archipelago and Little Belt are low relative to the total Kattegat-Skagerrak population, these colonies might become increasingly important in the future.

Based on non-parametric modelling, eleven out of twelve subregions in the Kattegat-Skagerrak displayed either a negative trend or little to no increase in mean count. The one subregion to display a positive trend, Marstrand, was bordered by the two subregions with the largest absolute rates of decline (Lysekil and Onsala) and may be influenced by local immigration. The exact delineation of colony boundaries within the Kattegat-Skagerrak is currently unknown. The subregion boundaries employed in the present study were based on assumed colonies used in previous work [4,14]. Many of these were defined based on geographic features, such as isolated islands or archipelagos harbouring key breeding sites (e.g., Anholt and Koster). For Sweden in particular, a few subregions also included large sections of coastline, which are unlikely to represent true colonies. Nonetheless, our analysis indicates that trends are largely consistent throughout the Kattegat-Skagerrak, with no single subregion driving apparent changes in abundance. Future work could aim to more clearly define colony boundaries to improve the assessment of localised trends in seal abundance [7,39].

We report relatively wide confidence intervals around estimated counts and rates of change. Harbour seal haul-out frequency is influenced by environmental factors such as temperature and wind speed [54–56]. Although surveys were conducted only on days with no precipitation and wind speeds of less than $10\,\text{ms}^{-1}$, at least some of the uncertainty we report likely results from differences in environmental conditions during surveys. Repetition of surveys in a single year and calculation of annual mean counts were conducted to minimise the impact of environmental variability on trend analysis [41]. A high level of uncertainty is common in wildlife monitoring and highlights the value of long-term monitoring programmes [57,58].

It is possible that the fraction of the population observed during the annual moult changed between 2003 and 2023. Competition between individuals for haul-out areas can occur and is likely to increase with population density, potentially leading to seals spending less time hauled out during peak moult and prolonging the moulting period [22,59,60]. Additionally, behavioural changes resulting from prey scarcity, such as increased time spent foraging, could lead to changes in haul-out frequency. The demographic structure of the hauled-out fraction of a harbour seal population changes throughout the moult due to differential behaviour of age- and sex-classes [61,62]. Changes in vital demographic rates can lead to a change in age structure, altering the fraction of the population hauled-out during the moult [62–64]. Disturbance of haul-out sites by vessels, and of seal foraging activities by offshore construction may also influence trends [65,66]. In 2020, there was a reported increase in recreational use of the coast by the Swedish public [67]. This may partially account for particularly low counts that year. Nonetheless, seals are required to spend time hauled-out during the annual moult to thermoregulate and promote growth of new fur and a large proportion of the population is expected to be constantly observed during surveys [68]. In future, targeted surveys using telemetry tags and/or individual identification methods should be used to study changes in haul-out behaviour relative to previous work [38,61]. Recent advances in drone-based survey methods could also be used to determine patterns in haul-out behaviour between age classes [19,69,70].

Globally, several harbour seal populations have experienced dramatic periods of decline in recent times [20,71], such as that in Sable Island, Canada [72]. It is unlikely that declining counts in the Kattegat-Skagerrak are an artefact of behavioural changes alone when considered in combination with independent data on localised reductions in pup counts and somatic growth in the Skagerrak [18,19]. Average annual adult survival in pinnipeds is generally high (often > 95% in harbour seals) with variation in population growth being primarily driven by birth rates and juvenile survival [13,73–75]. Given documented shifts in the Kattegat-Skagerrak ecosystem and the abundance of harbour seal prey species [26–28], it seems likely that food limitation, leading to lower birth rates, and potentially lower juvenile survival, is a contributing factor to declining counts. Such shifts in diet are likely driven by anthropogenic activities, such as overfishing, forestry,

and construction, which have led to widescale environmental degradation [26,27,35–37]. The effects of dietary changes could be investigated through comparative studies of the composition and nutritional value of diet and seal body condition between the Kattegat-Skagerrak and the growing S.W. Baltic population. Given the long history of ecological monitoring in the Kattegat-Skagerrak, there is also the opportunity to compare modern [34] and historical [29,30] diet composition.

Hunting by humans is a common cause of seal population decline, as evidenced by the collapse of the Kattegat-Skagerrak harbour seal population during the previous century and more recent declines in Icelandic waters [4,20]. In the Norwegian Skagerrak, between 35 and 49 harbour seals were taken annually in a quota hunt between 2015 and 2022 (data provided by the Directorate of Fisheries, Norway). In Sweden, protective hunting (allowing seals to be killed to protect fishing gear) has occurred since the early 2000s, and gradually increased to 390 individuals in 2020 [76,77]. In 2022, Sweden introduced a licensed hunt with a quota of approximately 700, although less than 50% of this was fulfilled [77]. Licensed hunting was not renewed in 2024 [76]. In 2018, Olsen et al. reported an average of 10 seals to be regulated through derogation shooting in Denmark [10] although this number has since increased. Given increases in the hunting of harbour seals in Sweden in the late 2010s [14,76,77], hunting is a possible contributing factor to reduced population growth in the Kattegat-Skagerrak [14]. It is unlikely, however, that hunting alone is responsible for reduced population growth given the negative trends observed in Danish colonies, where hunting pressure is negligible. Bycatch and entanglement in fishing gear can also impose limitations on seal population growth [5,21,78]. Levels of bycatch in the Kattegat-Skagerrak are poorly quantified for harbour seals [78,79]. At present, there is no reason to believe that bycatch levels have increased, and they may even be in decline as the use of passive fishing gear in the Kattegat-Skagerrak becomes less common [80,81].

Harbour seals are known to have been displaced by recovering grey seal (*Halichoerus grypus*) populations on multiple occasions [24,82]. However, grey seals remain uncommon in the Kattegat-Skagerrak [83,84]. Harbour seals are also prey to species such as white shark (*Carcharodon carcharias*) and killer whale (*Orcinus orca*) in parts of their range. White shark are not known to occur in the Kattegat-Skagerrak and the presence of killer whales is sporadic [25,85,86]. Interspecific interactions are therefore unlikely to be driving observed declines.

Declines in the growth of wildlife populations are often assumed to be the result of density dependence [14,87,88]. Here, intra-specific competition for resources causes a reduction in population growth as one or more resources become limiting [89,90]. This assumption is implicit in logistic growth models [43]. Harbour seals compete for resources such as haul-out sites [22,59] and potentially for prey [23]. Based on logistic modelling, we have estimated carrying capacities for the Kattegat-Skagerrak region (K = 13,965 ± 772 counted seals) as well as its subregions. These theoretical maximum values can be used in predictive modelling to assess the potential impact of management decisions on population viability [7,14,88]. Logistic models, however, assume a static ecosystem with a fixed carrying capacity [43]. A number of harbour seal populations around the world have experienced well documented shifts from growth to decline [20,71,72]. Exploring trends through more flexible non-parametric modelling is therefore of value, as a decline may indicate an environmental shift rather than the population reaching an equilibrium based on intra-specific competition.

The degree to which population growth is limited by intra-specific competition is a key question in estimating population viability and evaluating the sustainability of hunting quotas [14,91]. Reduced prey quality and abundance can limit population growth independently from predator density if nutritional needs are not being met [89,92,93]. During density dependent limitation, reductions in population size are expected to result in an increased growth rate [89,93,94]. In this case, mass mortality events, such as the 1988 and 2002 PDV outbreaks, are expected to result in a return to exponential growth and eventual recovery of the population if conditions remain favourable [12]. During density independent limitation however, lower growth rates will be maintained even if a large portion of the population is killed. This is true both for natural mortality events and for anthropogenic mortality, such as hunting. Thus, a population regulated by a deteriorating environment and not population density is at risk of overexploitation and should be managed in a precautionary manner

[91]. A mixture of density dependent and independent control of population growth is likely. Investigating the mechanisms by which population growth is regulated for the Kattegat-Skagerrak harbour seal population should now be considered a priority area of research.

## Conclusion

We have presented evidence of declining abundance of the Kattegat-Skagerrak harbour seal population. Declines are apparent at the regional level as well as locally for all but one subregion. The nearby population in the S.W. Baltic, in contrast, continues to increase (by approximately 50 animals per year). The changes in seal abundance reported here for the Kattegat-Skagerrak present an opportunity to study the mechanisms by which environmental factors regulate population growth and animal behaviour. While changes in haul-out behaviour may contribute to the apparent decline in counted harbour seals, it seems likely that nutrient limitation is also driving real shifts in seal abundance. Future work should investigate the role of factors such as migration, intra-specific competition, decreases in the abundance or quality of prey, availability of pupping sites, and hunting in determining population growth rates. Annual monitoring of harbour seal abundance represents a large investment of time and resources by national authorities. This effort is justified as marine mammal populations are indicators of overall environmental state. The decrease in seal abundance presented here provides an early-warning and should not be ignored by researchers or ecosystem managers. The management of harbour seals should be updated to account for a lower growth potential, for example by easing hunting pressure or improving seal sanctuary design to decrease the risk of further population decline.

## Supporting information

**S1 Fig. Parametric growth models.**
(PDF)

**S2 Fig. Rates of change for subregions.**
(PDF)

**S1 Table. Summary of Norwegian colony counts.**
(PDF)

**S2 Table. The outcomes of parametric modelling.**
(PDF)

**S3 Table. Estimates of maximum and 2023 counts.**
(PDF)

**S4 Table. Outcomes of GAM fitting for subregions.**
(PDF)

## Acknowledgments

The authors thank the pilots and observers who worked on the surveys as well as Dr. J. Harvey-Carroll for her feedback on the initial manuscript.

## Author contributions

**Conceptualization:** Daire Carroll, Karin C. Harding.

**Data curation:** Daire Carroll, Markus P. Ahola, Anja M. Carlsson, Anders Galatius, Kjell T. Nilssen.

**Formal analysis:** Daire Carroll.

**Investigation:** Markus P. Ahola, Anja M. Carlsson, Anders Galatius, Kjell T. Nilssen.

**Writing – original draft:** Daire Carroll.

**Writing – review & editing:** Daire Carroll, Markus P. Ahola, Anja M. Carlsson, Anders Galatius, Kjell T. Nilssen, Tero Härkönen, Karin C. Harding.

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
