## [Decision Letter · Decision Letter 0]

15 Apr 2025

PONE-D-25-11452Declining harbour seal abundance in the Kattegat-Skagerrak and Danish StraitsPLOS ONE

Dear Dr. Carroll,

Thank you for submitting your manuscript to PLOS ONE. After careful consideration, we feel that it has merit but does not fully meet PLOS ONE’s publication criteria as it currently stands. Therefore, we invite you to submit a revised version of the manuscript that addresses all the points raised during the review process, particularly the methodological issues.  

We look forward to receiving your revised manuscript.

Kind regards,

Aldo Corriero, Ph.D.

Academic Editor

PLOS ONE

 [Swedish surveys were funded by the Swedish Environmental Protection Agency, and the Swedish Agency for Marine and Water Management (no specific grant codes avalible). Danish surveys were funded by the Danish Environmental Protection Agency (no specific grant codes avalible). DC was funded by the Wild Animal Initiative Fellowship, grant numbers F-2023-00005 and the Swedish Environmental Protection Agency through the Environmental research fund and the Swedish Research Council Formas, grant number (2024-00147). KCH was supported by the European Union (HORIZON-CL6-2023-BIODIV-01-101135307).].

6. We note that Figure 1 in your submission contain [map/satellite] images which may be copyrighted. All PLOS content is published under the Creative Commons Attribution License (CC BY 4.0), which means that the manuscript, images, and Supporting Information files will be freely available online, and any third party is permitted to access, download, copy, distribute, and use these materials in any way, even commercially, with proper attribution. For these reasons, we cannot publish previously copyrighted maps or satellite images created using proprietary data, such as Google software (Google Maps, Street View, and Earth). For more information, see our copyright guidelines: http://journals.plos.org/plosone/s/licenses-and-copyright.

Additional Editor Comments (if provided):

Reviewers' comments:

Reviewer's Responses to Questions

**Comments to the Author**

1. Is the manuscript technically sound, and do the data support the conclusions?

Reviewer #1: Yes

Reviewer #2: Partly

2. Has the statistical analysis been performed appropriately and rigorously? 

Reviewer #1: Yes

Reviewer #2: No

3. Have the authors made all data underlying the findings in their manuscript fully available?

Reviewer #1: Yes

Reviewer #2: Yes

4. Is the manuscript presented in an intelligible fashion and written in standard English?

Reviewer #1: Yes

Reviewer #2: Yes

5. Review Comments to the Author

Reviewer #1: This study provides a well-structured and comprehensive analysis of the decline in harbour seal abundance in the Kattegat-Skagerrak and Danish Straits. The authors employ a robust study design, utilizing long-term aerial survey data (2003–2023) and applying both parametric and non-parametric modeling approaches to assess trends. The use of logistic and exponential growth models, alongside Generalized Additive Models (GAMs), effectively captures the temporal patterns and regional differences in population dynamics. The manuscript is well-written, and the analysis is straightforward, making the results accessible and easy to interpret.

The results indicate an annual decline of 408 individuals in the Kattegat-Skagerrak, but the confidence interval is relatively wide (95% CI: [67, -882]). While this reflects uncertainty in the estimates, a brief discussion of the implications of this variability in trend estimates would strengthen the interpretation. Some reorganization of Introduction might be helpful to streamline the manuscript. It is recommended to check the text for consistency of terminology used and avoid repetitive expansion of acronyms.

The suggested revisions provided in the text primarily focus on minor clarifications and additional context to further strengthen the paper.

Reviewer #2: This represents a nice summary of harbor seal trends in a lesser well represented area of the world. However, I felt that the explanation of statistical methods needed more detail, specifically in terms of accounting for changes in survey effort over time, and what models were being compared with AIC.

6. PLOS authors have the option to publish the peer review history of their article (what does this mean? ). If published, this will include your full peer review and any attached files.

**Do you want your identity to be public for this peer review?** For information about this choice, including consent withdrawal, please see our Privacy Policy .

Reviewer #1: No

Reviewer #2: No

---

## [Author Response · Author response to Decision Letter 1]

22 May 2025

We provide detailed responses to all editorial and reviewer comments in the file "Revision 2 Responses.docx".

---

## [Decision Letter · Decision Letter 1]

9 June 2025

Declining harbour seal abundance in a previously recovering meta-population

PONE-D-25-11452R1

Dear Dr. Carroll,

We’re pleased to inform you that your manuscript has been judged scientifically suitable for publication and will be formally accepted for publication once it meets all outstanding technical requirements.

Kind regards,

Aldo Corriero, Ph.D.

Academic Editor

PLOS ONE

Additional Editor Comments (optional):

All the reviewers' comments have been been addressed and the manuscript can be accepted for publication.

Reviewers' comments:

Reviewer's Responses to Questions

**Comments to the Author**

1. If the authors have adequately addressed your comments raised in a previous round of review and you feel that this manuscript is now acceptable for publication, you may indicate that here to bypass the “Comments to the Author” section, enter your conflict of interest statement in the “Confidential to Editor” section, and submit your "Accept" recommendation.

Reviewer #2: All comments have been addressed

2. Is the manuscript technically sound, and do the data support the conclusions?

Reviewer #2: Yes

3. Has the statistical analysis been performed appropriately and rigorously? 

Reviewer #2: Yes

4. Have the authors made all data underlying the findings in their manuscript fully available?

Reviewer #2: Yes

5. Is the manuscript presented in an intelligible fashion and written in standard English?

Reviewer #2: Yes

6. Review Comments to the Author

Reviewer #2: My comments were thoughtfully addressed in the response to reviewer comments. The authors present a sound analysis and a thoughtful discussion of the results.

7. PLOS authors have the option to publish the peer review history of their article (what does this mean? ). If published, this will include your full peer review and any attached files.

**Do you want your identity to be public for this peer review?** For information about this choice, including consent withdrawal, please see our Privacy Policy .

Reviewer #2: No

---

## [Editor Report · Acceptance letter]

PONE-D-25-11452R1

PLOS ONE

Dear Dr. Carroll,

I'm pleased to inform you that your manuscript has been deemed suitable for publication in PLOS ONE. Congratulations! Your manuscript is now being handed over to our production team.

Kind regards,

on behalf of

Dr. Aldo Corriero

Academic Editor

PLOS ONE